# Stateful characterization of resistive switching TiO$_2$ with electron beam induced currents

Brian D. Hoskins[1,2], Gina C. Adam[3,4], Evgheni Strelcov[1,5], Nikolai Zhitenev[1], Andrei Kolmakov [1], Dmitri B. Strukov[3] & Jabez J. McClelland [1]

Metal oxide resistive switches are increasingly important as possible artificial synapses in next-generation neuromorphic networks. Nevertheless, there is still no codified set of tools for studying properties of the devices. To this end, we demonstrate electron beam-induced current measurements as a powerful method to monitor the development of local resistive switching in TiO$_2$-based devices. By comparing beam energy-dependent electron beam-induced currents with Monte Carlo simulations of the energy absorption in different device layers, it is possible to deconstruct the origins of filament image formation and relate this to both morphological changes and the state of the switch. By clarifying the contrast mechanisms in electron beam-induced current microscopy, it is possible to gain new insights into the scaling of the resistive switching phenomenon and observe the formation of a current leakage region around the switching filament. Additionally, analysis of symmetric device structures reveals propagating polarization domains.

[1] Center for Nanoscale Science and Technology, National Institute of Standards and Technology, Gaithersburg, MD 20899, USA. [2] Materials Department, University of California Santa Barbara, Santa Barbara, CA 93106, USA. [3] Electrical and Computer Engineering Department, University of California Santa Barbara, Santa Barbara, CA 93106, USA. [4] Institute for Research and Development in Microtechnologies, 077190 Bucharest, Romania. [5] Institute for Research in Electronics and Applied Physics, University of Maryland, College Park, MD 20742, USA. Correspondence and requests for materials should be addressed to B.D.H. (email: brian.hoskins@nist.gov)

Metal oxide resistive switches (also known as ReRAM or memristors) have been of intense interest for use in next-generation memory or as analog weights in neuromorphic networks[1–3]. Their unique properties, including two-terminal structure, scalability (more than 10 nm), nonvolatility (more than 10 years at 85 °C), high endurance (more than $10^{12}$ cycles), and low energy consumption (less than 10 pJ), are ideal for next-generation hardware[4–6]. However, the complex nature of the switching in these devices, speculated to involve coupling of chemical, electrical, and thermal fields, has stymied a comprehensive understanding of the process[7].

A metal oxide resistive switch consists of two metallic layers separated by a substoichiometric oxide and acts as a programmable resistor. While the switching process is not fully understood, it is generally believed to involve the motion of oxygen vacancies and metal cations in the oxide under electrical fields and thermal gradients[8–10]. This ion motion leads to a local, nanometer-scale variation in the vacancy concentration and a corresponding variation in the thickness of the oxide's depletion region. The vacancy concentration often appears to grow like a metallic, nanoscale filament, and as the depletion region thickness declines, the conductance passes from being controlled by thermionic emission, to being controlled by thermionic field emission, and ultimately to being controlled by field emission.

Several studies have been conducted probing the underlying physics of the switching and exploring the origin and dynamics of filament formation in resistive switches. However, work so far has not yielded a comprehensive picture. Transmission electron microscopy studies suggest that the underlying structural changes during switching can be small, particularly under conditions where the switching is controlled by current compliance[11]. Investigations using scanning transmission X-ray microscopy have probed the chemical changes in devices through the forming process, but have not managed to view single cycle changes or correlate observed large area chemical changes with local changes in the conductivity[12,13]. The chemical variations between conductive and insulating configurations are highly localized and are therefore hard to quantify with available spectroscopic tools, particularly in the presence of a large deformation region[14]. Measurements with scalpel scanning probe microscopy (SPM) have provided precise measurements of filament behavior. However, the destructive nature of this technique makes it difficult to explore the full parameter space, since it only analyzes a single switching event[15].

In the present work, we employ high-resolution scanning electron microscopy (SEM)-based electron beam-induced current microscopy (EBIC) to systematically explore switching in $TiO_2$ devices such as in Fig. 1a, b. We present detailed imaging of filament formation as a function of resistive state (see Fig. 1c, d),

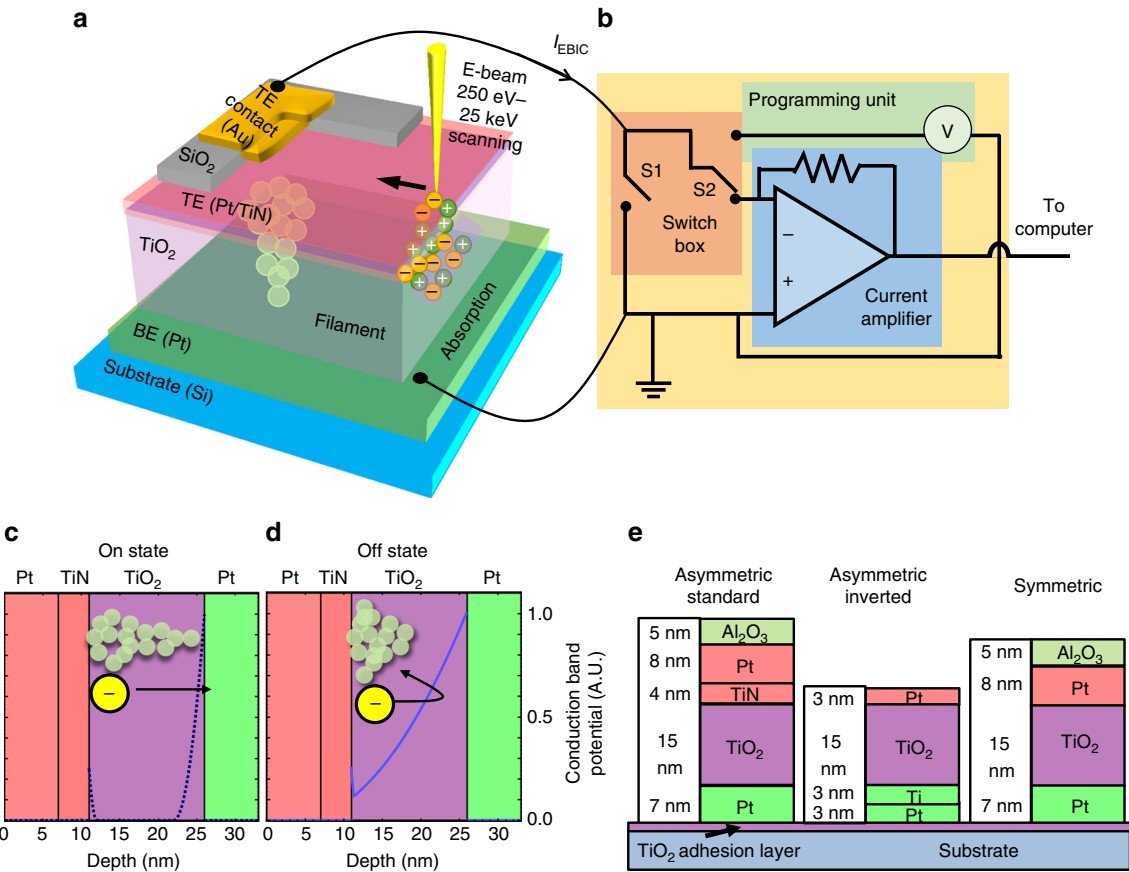

**Fig. 1** Overview of experiment. **a** 3D depiction of the experimental EBIC measurement including top electrode (TE, pink), the TE contact (yellow), bottom electrode (BE, green), e-beam with its generated carriers, dielectric layer ($TiO_2$, purple), and the programmable filament. **b** Basic electric measurement setup including switch box with grounding switch (S1) and exchange between imaging (with the current amplifier) and programming (S2). **c** Simplified schematic depiction of the filament in the on state (spanning the top electrode to the bottom electrode) and **d** the off state (leaving an insulating barrier). **e** Device stack of the different structures analyzed. The protective aluminum oxide was stripped from the inverted device and the Pt made thinner in an attempt to increase the resolution

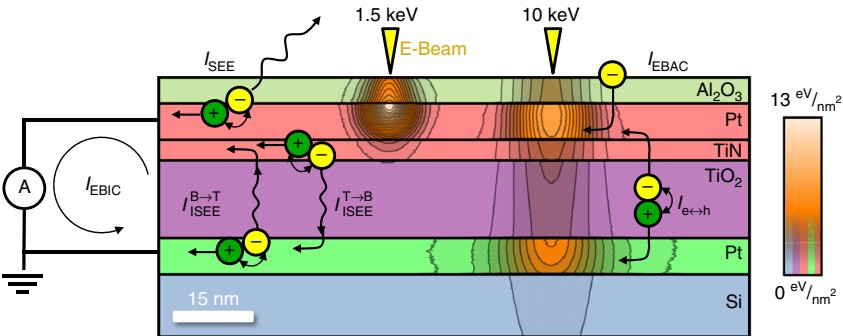

**Fig. 2** Depiction of device beam interaction. Monte Carlo-simulated absorption in a multi-layer ReRAM device at both 1.5 keV incident beam and 10 keV incident beam. Absorption in different layers can result in the different depicted currents including the secondary electron current ($I_{SEE}$), the electron beam absorbed current ($I_{EBAC}$), the electron–hole pair current ($I_{e\leftrightarrow h}$), and the internal secondary electron currents from top to bottom ($I_{ISEE}^{T\rightarrow B}$) and bottom to top ($I_{ISEE}^{B\rightarrow T}$). These all sum to create the measured electron beam-induced current ($I_{EBIC}$). Energy absorption scale bar indicates high intensity (white), intermediate intensity (orange), and zero intensity (transparent revealing diagram coloring of corresponding device layers)

and also explore the physical mechanisms of current generation. EBIC has been used previously to characterize resistive switching devices, but relatively low resolution (~1 μm) and the absence of a fundamental understanding of the mechanisms of image formation—due to a lack of stateful and energy-dependent data—has so far made it impossible to draw conclusions as to the origin of the generated current and the nature of the filament and its surrounding deformation region[16,17]. By exploring variations in the generated signal in different device geometries (Fig. 1e) as a function of beam energy and resistive state, we show it is possible to probe the underlying physics of the resistive switching devices, clarify the image formation mechanisms, and develop a reliable means of observing filament formation and distinguishing it from non-filament areas. This approach to stateful characterization is robust to measurement artifacts by being selective to the reversible changes in the device and only those regions electrically connected to the circuitry. To provide the most possible information, three different kinds of metal oxide resistive switches (Fig. 1e) were constructed: asymmetric structures were made in a standard form, Pt/TiO$_x$/TiN/Pt, and an inverted form, Pt/Ti/TiO$_x$/Pt, and a symmetric form, Pt/TiO$_x$/Pt, was also made.

In EBIC, few electron–volt secondary electrons and electron–hole pairs created by a primary beam at 250 eV–25 keV (Fig. 1a) interact strongly with the built-in device fields. Current is collected via auxiliary electrodes connected to different parts of the device (Fig. 1b), resulting in a local measurement of the electronic structure stimulated by the incident electron beam[18].

The large number of interfaces in a metal–insulator–metal (MIM) structure such as a metal oxide resistive switch leads to competing currents that sum to the measured EBIC current ($I_{EBIC}$). We can write:

$$I_{EBIC} = I_{EBAC} + I_{SEE} + I_{(e\leftrightarrow h)} + I_{ISEE}^{T\rightarrow B} + I_{ISEE}^{B\rightarrow T}, \quad (1)$$

where $I_{EBAC}$ is the current absorbed from the incident electron beam (incident current less any backscattered or transmitted current), $I_{SEE}$ is the secondary electron emission current, $I_{e\leftrightarrow h}$ is the electron–hole pair separation current, $I_{ISEE}^{T\rightarrow B}$ is the internal secondary electron emission current from the top electrode to the bottom electrode, and $I_{ISEE}^{B\rightarrow T}$ is the internal secondary electron emission current from the bottom electrode to the top electrode. Figure 2 shows approximate locations of the sources of these currents and their polarities. $I_{EBAC}$ and $I_{SEE}$ are universal to all materials, since they represent the injected current and emitted electrons

to vacuum. $I_{e\leftrightarrow h}$, produced when e-beam-induced electron–hole pairs are created at junctions between materials and separated by built-in fields, is often the largest and most commonly measured EBIC partition[19–21]. $I_{ISEE}^{T\rightarrow B}$ and $I_{ISEE}^{B\rightarrow T}$ are unique to MIM structures and result from thermionic emission, diffusion, or tunneling of hot electrons from one electrode to the other[22–24]. In a conventional MIM diode, the internal secondary electron currents are usually negligible, but become measurable at large applied biases ($V_b$) due to a lowering of the effective barrier height.

Each current can be a probe of the device behavior, e.g., indicating morphological changes such as crystallization or coarsening. Since barrier lowering and raising is resistive switching, measurable quantities of $I_{ISEE}^{T\rightarrow B}$ and $I_{ISEE}^{B\rightarrow T}$ are observable in the absence of an applied bias when the device is switched to the on state, increasing the probability of hot electron transmission.

It can be difficult to deconvolve the sources of current in the device. However, a general principle of EBIC is that the observed current and electron yield, $Y_{EBIC}$ (or nanoampere of signal per nanoampere of injected current), is proportional to the energy deposited into the specific layer sourcing the current. Since beam penetration and absorption primarily depend on the incident beam energy, the ratios of different current contributions will likewise depend on the incident beam energy[25]. Variations in these ratios are predicted by simulating the energy absorbed using Monte Carlo electron simulators[26–28]. Figure 2 shows a two-dimensional projection, and Fig. 3a shows a 1-d depth profile, for these processes in a Pt/TiO$_x$/TiN/Pt structure for different energies. Integrating all of the energy between two depths as a function of beam voltage makes it possible to define energy absorption functions for different regions of the device $f_{absorbed}^{Top\ electrode}$ (Fig. 3b), $f_{absorbed}^{TiO_2}$ (Fig. 3c), and $f_{absorbed}^{Bottom\ electrode}$ (Fig. 3d), which simulate the amount of energy absorbed in each device layer per electron. These can be used to predict currents sourced from those layers, such as $I_{ISEE}^{T\rightarrow B}$, $I_{e\leftrightarrow h}$, and $I_{ISEE}^{B\rightarrow T}$, or their normalized yields, respectively.

Here we will show that it is possible to faithfully reproduce the EBIC energy dependence of a ReRAM device as a function of state, using only the independently measured fabrication parameters of the device, the independently modeled e-beam–matter interaction, and physical intuition of a device's internal electrical fields and resistance. Collectively these can be used to deconstruct complicated effects such as hot electron transmission, material recrystallization, parasitic leakage, and built-in field inversion.

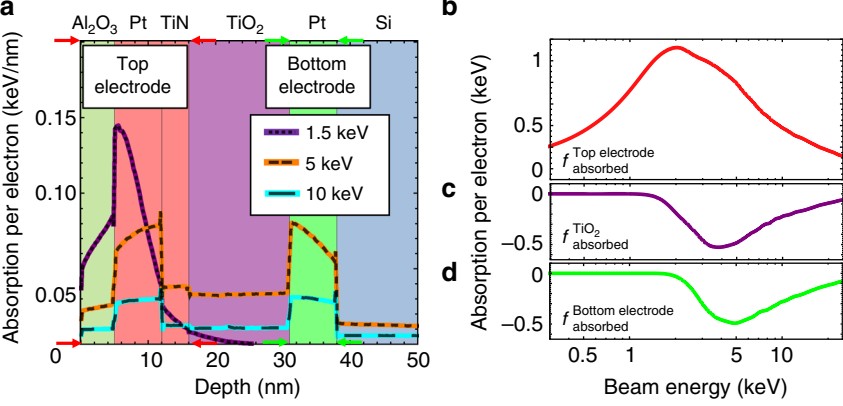

**Fig. 3** Distribution of energy in device. **a** Simulated one-dimensional absorption profile in an ReRAM stack for different beam energies. Energy absorbed in the TiO₂ layer first rises and then falls with increasing beam energy. Note any energy absorbed in the silicon does not contribute to the measured electron beam-induced current. Arrows indicate regions of energy over which top and bottom electrodes are integrated over for the layer-by-layer energy distributions. **b–d** Monte Carlo simulations on an asymmetric standard structure of absorption in **b** top layer, **c** TiO₂ layer, and **d** bottom layer as a function of beam energy summarizing contributions from all elements of the structure

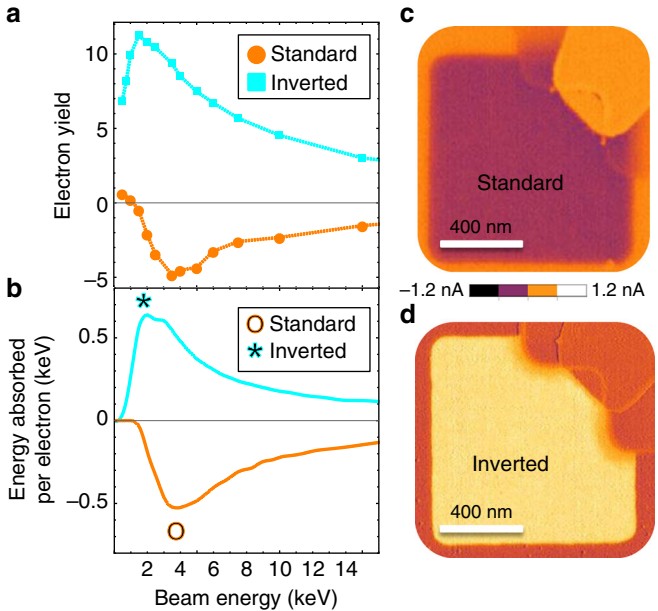

**Fig. 4** Pristine device measurements. **a** Measured energy dependence of the electron beam-induced current (EBIC) signal for the virgin device in standard and inverted devices. The measured current is negative in the standard device and positive in inverted device. The change in sign when flipping the device over is due to reversal of the built-in field. Error bars, reflecting two standard deviations of the mean, are smaller than the markers and are determined from an area of at least 50 × 50 pixel bounding box on each pad. **b** Monte Carlo simulation results of inverted and standard structure absorbed energy per electron. Standard structure simulation result is shown with negative sign since absorbed energy is not a negative quantity and current direction is determined by structure. **c** 5 keV EBIC image of a standard device. **d** 2 keV EBIC image of an inverted device. The strong similarity between the measured and simulated curves show the strong relationship between absorbed energy and generated EBIC current

## Results

**Virgin device measurements.** In measurements of virgin asymmetric devices, the EBIC signal arises from an electron flow from low work function (TiN) to high work function contacts (Pt). Consequently, standard device structures exhibited negative

absolute current, and inverted structures exhibited positive absolute current (Fig. 4a). Symmetric structures were observed to have a more complex behavior, occasionally exhibiting one polarity or the other as well as significant relaxation and charging of the pad (see Supplementary Note 4).

The energy dependence of the EBIC signal for the different device structures followed the Monte Carlo simulations of energy absorbed in the TiOₓ, achieving maximum amplitude at 3.5 keV and 2 keV for the standard and inverted structures, respectively (Fig. 4b). For the standard structure (a top TiN layer), the signal polarity switches from positive to negative at energies above 1 keV, as the secondary electron emission into the vacuum (at low energy) is overcome by the background hole pair signal (note that the secondary electron emission signal is not included in the Monte Carlo model)[29]. EBIC micrographs of the devices showed strong contrast from the surrounding isolation and electrodes (Fig. 4c, d).

The internal quantum efficiency of the EBIC process can be estimated by combining the simulations with the Alig and Bloom relation[30,31]:

$$E_i \approx 3E_g + 1\text{eV}, \tag{2}$$

where $E_i$ (the effective pair creation energy) is substituted for $E_g$ (the band gap) in calculations for collection efficiency, $\eta$, such that:

$$\eta \approx \frac{E_i}{E_{absorbed}} \frac{I_{EBIC}}{I_{beam}} = \frac{E_i}{E_{absorbed}} Y_{EBIC}, \tag{3}$$

where $I_{EBIC}$ and $I_{beam}$ are the measured EBIC and incident beam currents, respectively, and $Y_{EBIC}$ is the resulting yield from their ratio[32]. For the standard structure at 5 keV, the simulation predicts that ~50 electron–hole pairs per incident electron should be generated, as opposed to an approximately measured four electron–hole pairs, suggesting a collection efficiency of 8%.

**Switched device, asymmetric structure.** Switched-on devices exhibited morphological changes including electrode changes, both minor and severe, visible by the secondary electron detector, as well as morphological changes in TiO₂ visible in the EBIC signal. Minor changes in the electrode include grain coarsening, which was easily visible as increased secondary electron emission (or image brightening) relative to the unaffected areas[33–36]. More severe changes include tearing of the electrode. For measurements

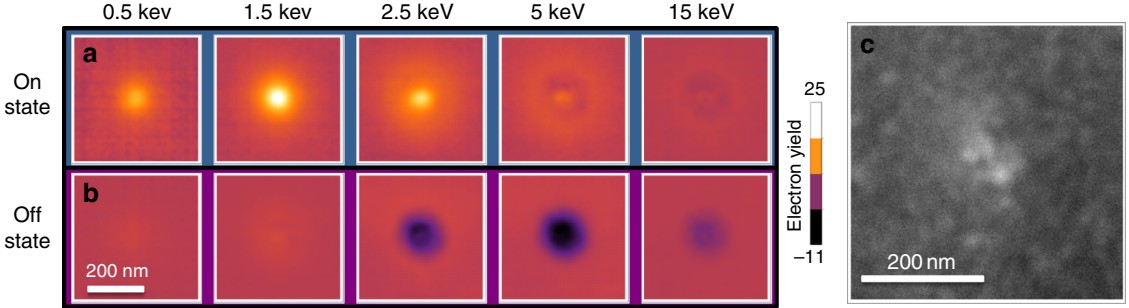

**Fig. 5** Micrographs of filament. **a** Electron beam-induced current micrograph series showing contrast evolution with beam energy for the on state and **b** for the off state. The on-state signal maximum implies the signal is due to absorption in the top electrode, whereas the the off-state signal minimum implies the signal is due to absorption in the TiO₂ layer. **c** Scanning electron micrograph of the device after switching showing no tears in the electrode

potentially sensitive to changes in the electrode, devices with torn electrodes were not used.

In what follows, the signal discussed refers to the change in EBIC current relative to the current measured in virgin structures (the background current). Energy-dependent beam measurements of the device in both the off and the on state (Fig. 5) show a strong dependence on the device resistivity in both the polarity and magnitude of the signals. In the off state, the formed region shows a region of enhanced dark contrast, which could be as small as 100 nm in diameter to as large as 300 nm. Its signal maximum, between 2.5 and 5 keV, suggests it is due to an enhanced electron–hole pair current, $I_{e \leftrightarrow h}$, arising from crystallization of the oxide, which is known to be a consequence of forming resistive switches[11,17,36].

Within this broader darker region, which we will call the crystallized region, a new, positive signal appears at low beam energies in the on state. This positive signal correlates with the state of the device, and vanishes when the device is programmed into the off state—indicating the signal is associated with the filament. Measurements as a function of energy showed that the signal achieves a maximum at 1.5 keV, a value consistent with top electrode absorption (Fig. 3b). The polarity and beam energy of the signal maximum suggest that the EBIC signal is due to internal secondary electron emission (ISEE). This is further supported by determining whether the polarity of the filament signal remains unchanged in the inverted devices. Since beam-electrode collisions generate hot electrons, the signal current polarity should be independent of the filament orientation, and, indeed, inverting the device did not cause a reversal in the polarity of the signal (see Supplementary Note 6).

A clearer picture of the different current contributions as a function of state can be obtained by doing azimuthal integration to average around the filament location and then plotting the radial current distribution. Figure 6a shows the off-state distribution with a bottom plateau and a broad, ill-defined edge region, where $I_{e \leftrightarrow h}$ declines monotonically to the background, possibly corresponding to a transition between polycrystalline and the surrounding amorphous regions. Figure 6b, c shows the change in the on-state profiles with increasing beam energy. The plot of the internal secondary electron emission from the top electrode to the bottom electrode $I_{ISEE}^{T \rightarrow B}$ shows significant broadening from the diffusion hot of electrons across the top electrode to the filament. At 5 keV, it is significantly mixed with both the internal secondary electron emission from the bottom electrode to the top electrode, $I_{ISEE}^{B \rightarrow T}$, and the electron–hole pair separation current, $I_{e \leftrightarrow h}$, causing oscillations to begin emerging. At the highest beam energies, $I_{ISEE}^{B \rightarrow T}$ and $I_{e \leftrightarrow h}$ exhibit a substantially higher fraction of the total overall signal and the narrower diffusion widths and smaller magnitudes of $I_{ISEE}^{B \rightarrow T}$ and $I_{e \leftrightarrow h}$ create an oscillatory cross-section of the EBIC profile.

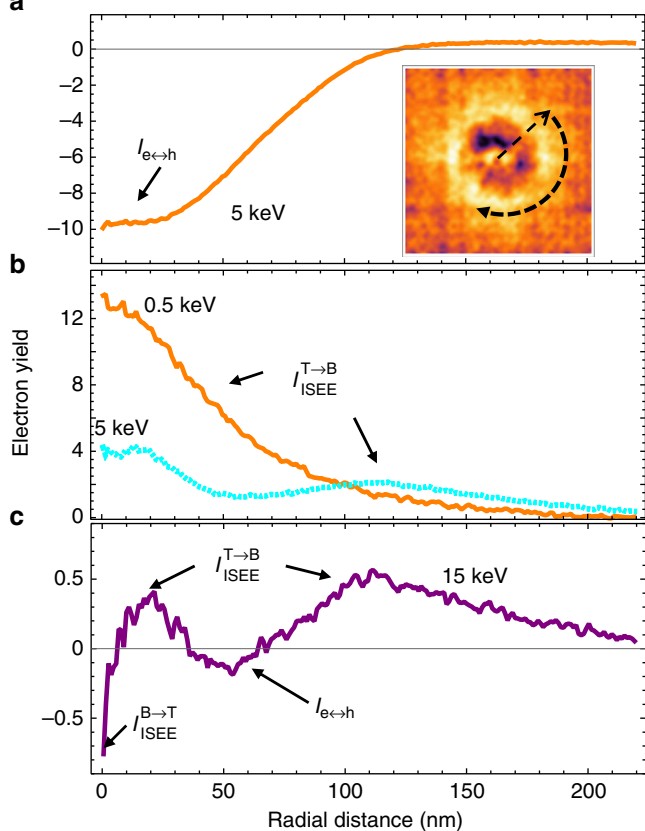

**Fig. 6** Radial distributions of EBIC. **a** Radial plot of the azimuthally averaged electron beam-induced current (EBIC) electron yield for a device in the off-state measured at 5 keV with the current mostly attributed to electron–hole pair separation and (inset) pictorial depiction of azimuthal averaging of an EBIC map. **b** Radial plot of the EBIC electron yield in the on state at low energy (0.5 keV) and an intermediate energy (5 keV) showing a signal dominated by ISEE from top to bottom but with an increasing contribution from the other currents. **c** Radial plot in the EBIC electron yield in the on state at 15 keV showing different regions with different dominant currents (indicated by current label and arrows) leading to a radially oscillating EBIC profile

The energy dependencies of the signals can be more carefully considered by using the Monte Carlo modeled energy absorption functions ($f_{absorbed}^{Top electrode}$, $f_{absorbed}^{Bottom electrode}$, $f_{absorbed}^{TiO_2}$) to model the electron yields ($Y_{ISEE,On(Off)}^{T \rightarrow B}$, $Y_{ISEE,On(Off)}^{B \rightarrow T}$, $Y_{e \leftrightarrow h}$) as a function of energy. Each of the absorption functions $f_{absorbed}$ describes the incident energy dependence of all of the energy absorbed in each

layer of the device, as well as the expected sign of the resultant EBIC current. Assuming proportionality of the yields to the absorption functions, we write:

$$Y_{\text{ISEE,On(Off)}}^{\text{T}\rightarrow\text{B}} = a_{\text{On(Off)}} f_{\text{absorbed}}^{\text{Top electrode}} \tag{4}$$

$$Y_{\text{ISEE,On(Off)}}^{\text{B}\rightarrow\text{T}} = a_{\text{On(Off)}} \delta f_{\text{absorbed}}^{\text{Bottom electrode}} \tag{5}$$

$$Y_{\text{e}\leftrightarrow\text{h}} = c f_{\text{absorbed}}^{\text{TiO}_2} \tag{6}$$

The constants of proportionality for the ISEE signals $a_{\text{On(Off)}}$ have a different value depending on whether the device is in the on or off state. The factor $\delta$ is a constant representing the relative yields between $Y_{\text{ISEE}}^{\text{T}\rightarrow\text{B}}$ and $Y_{\text{ISEE}}^{\text{B}\rightarrow\text{T}}$, and $c$ is a constant describing the electron–hole pair background signal for $Y_{\text{e}\leftrightarrow\text{h}}$. Both $c$ and $\delta$ are taken to be state independent. The total EBIC yield $Y_{\text{EBIC}}^{\text{On(Off)}} = Y_{\text{ISEE,On(Off)}}^{\text{T}\rightarrow\text{B}} + Y_{\text{ISEE,On(Off)}}^{\text{B}\rightarrow\text{T}} + Y_{\text{e}\leftrightarrow\text{h}}$ can then be written for the on and off states as:

$$Y_{\text{EBIC}}^{\text{On}} = a_{\text{On}} \left( f_{\text{absorbed}}^{\text{Top electrode}} + \delta f_{\text{absorbed}}^{\text{Bottom electrode}} \right) + c f_{\text{absorbed}}^{\text{TiO}_2} \tag{7}$$

$$Y_{\text{EBIC}}^{\text{Off}} = a_{\text{Off}} \left( f_{\text{absorbed}}^{\text{Top electrode}} + \delta f_{\text{absorbed}}^{\text{Bottom electrode}} \right) + c f_{\text{absorbed}}^{\text{TiO}_2} \tag{8}$$

We then allow the four coefficients $a_{\text{On}}, a_{\text{Off}}, \delta$, and $c$ to be free parameters in a simultaneous least-squares fit of both Eqs. 7 and 8 to the measured electron yields for the on and the off state. The result of the fit is shown in Fig. 7.

It is apparent from Fig. 7 that a simple linear combination model with only four free parameters can reproduce the most important features of the on-state and off-state EBIC curves, including the locations of the maximum, minimum, and the decay at high-incident beam energy, with only the coefficient of the ISEE signal changing between the states. This is done fairly accurately using only the independently measured film

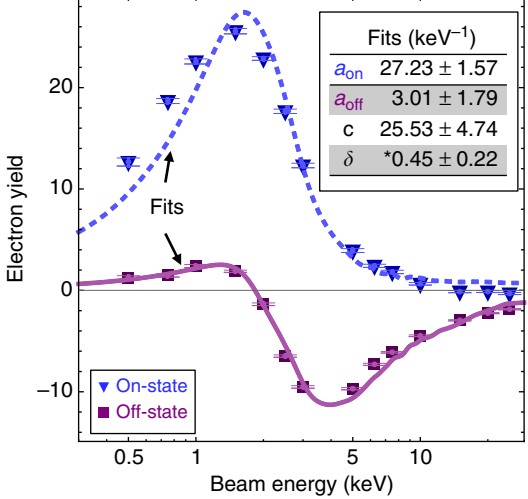

**Fig. 7** Experimental fit of beam energy dependence. Measured electron yield as a function of energy. Error bars indicate two standard deviations of the mean within an 11 × 11 pixel box at the signal center for the on-state and off-state micrograph series in Fig. 5. All quantities are measured with respect to the background. Fits are shown with least-squares 95% confidence intervals with units of keV$^{-1}$ (*note $\delta$ is dimensionless). Only the least-squares coefficient for the internal secondary electron emission (ISEE) signal, $a$, with associated variables $a_{\text{on}}$ and $a_{\text{off}}$ changes between the off state and the on state

thicknesses from the fabrication as input to the Monte Carlo model, without resorting to adjustments for tilt or density. This simplistic model does surprisingly well, considering the underlying complexity of the processes involved. A more detailed model would likely have to include the details of hot electron transport in diodes in addition to the energy absorption[22,23].

**Scaling of the internal secondary electron emission signal.** Tracking the ISEE signal, and by proxy the coefficient $a$, across a region provides both a qualitative (see Supplementary Movie 1) and a quantitative measure of the barrier to hot electron conductance and its scaling with device resistance. This can be done by continuously tuning the device resistance and measuring the change in total signal at a single beam energy. Summing up the total differential current with respect to the off-state through a turn-on and turn-off event, it is apparent that the ISEE signal follows a power law with exponent less than 1 as a function of conductance as the device is programmed into the off state (Fig. 8a), and scales nearly linearly with conductance as the device is programmed into the on state (Fig. 8b). The different scaling relationships between the turn-on and turn-off branches suggest contrasting mechanisms for the filament formation and dissolution, respectively.

One interpretation suggests that the on-branch switching is area dependent, driven by nucleation, saturation, and expansion of the filament, producing a signal proportional to the area ($A$), whereas the turn-off process is barrier dependent (through $\varphi_{\text{eff}}$, an effective barrier to conductance), and therefore determined entirely by a local state variable (such as the oxygen vacancy concentration). Such a difference has been proposed in some thermophoresis-based models of resistive switching[37,38]. In drift-diffusion models, it is potentially possible to explain based on the differing dopant profiles produced by drift/diffusion acting in concert or in opposition to one another, as well as by including two-dimensional effects[37,39].

In filaments models, competing modes of conduction often include Poole–Frenkel emission, space charge-limited conduction, and interfacial resistance depending on the resistive state[40,41]. These models capture the most attractive feature of resistive switches, i.e., the ability to continuously tune their resistance by varying some effective barrier height, $\varphi_{\text{eff}}$, between maximal ($\varphi_{\text{off}}$) and minimal ($\varphi_{\text{on}}$) values. In the case of TiO$_2$, the device resistance is often modeled as being controlled by an interfacial Schottky junction with a variable Arrhenius factor controlled by an exponent, $\frac{\varphi_{\text{eff}}}{k_B T_o}$,[42–45]. This approach is also sometimes used for other systems, especially SrTiO$_3$[46–48]. While it can be fairly accurate for the off state, its accuracy will decline in the limit of high conductivity as field emission dominates the transport across the interface and the conductance becomes limited by the quantum or Sharvin point contact resistance[49]. In this case, the temperature dependence could be expected to change from insulating to metallic[50]. However, it will be used here as a first approximation of the relationship between the conductance and the ISEE signal.

The scaling relationships during the turn-on and turn-off branches can be analyzed assuming the zero-bias conductance $\sigma$ follows the behavior of a Schottky junction,

$$\sigma = \left.\frac{dI_{\text{Schottky}}}{dV}\right|_{(V=0)} = \alpha A e^{-\frac{\varphi_{\text{eff}}}{k_B T_o}}. \tag{9}$$

where $\alpha$ is a prefactor, $A$ is the area of the filament, $\varphi_{\text{eff}}$ is the effective barrier height, and $T_0$ is the ambient temperature[51]. The ISEE signal can be assumed to follow thermionic emission theory, with a characteristic hot electron temperature, $T_e$, such

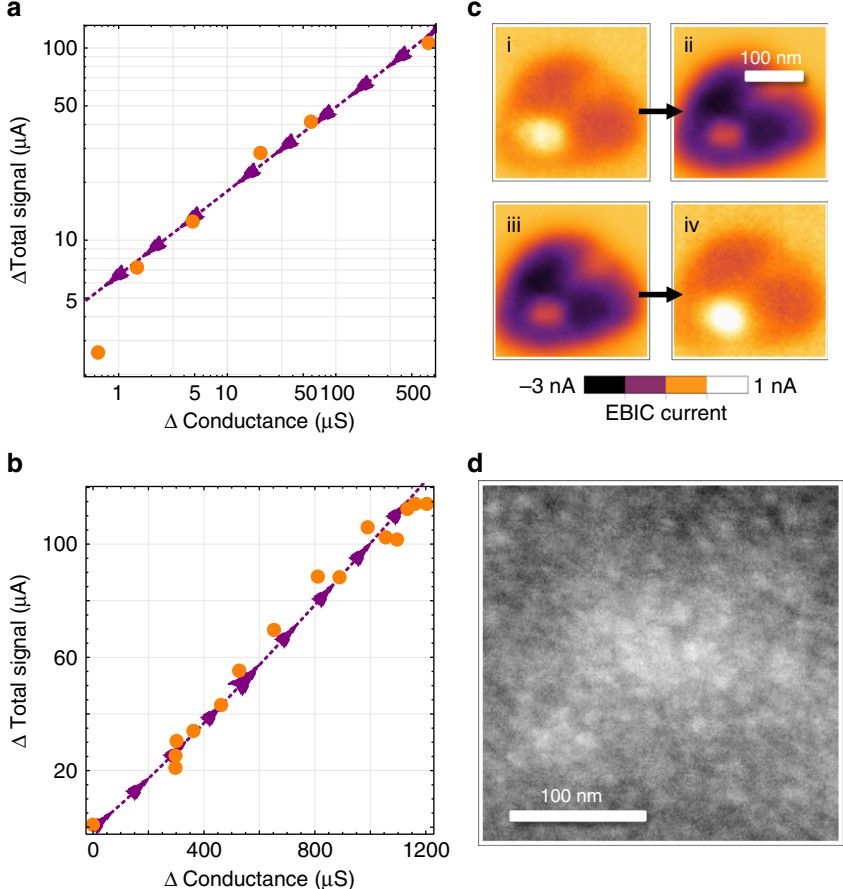

**Fig. 8** Conductance signal relationships. **a** Comparison of the change in conductance measured at 0.1 V from the off state (ΔConductance) and the measured change in total signal in an image (ΔTotal signal). Scaling dependence of the total electron beam-induced current (EBIC) signal at 5 keV in turn-off branch with sublinear exponent 0.43 ± 0.15 (95% confidence). **b** Scaling dependence of total EBIC signal in turn-on branch with near linear exponent 1.10 ± 0.10 (95% confidence), **c** EBIC depiction of an on (i) to of (ii) and off (iii) to on (iv) transition. A compilation of switching micrographs for this process is available in Supplementary Movie 1. **d** Scanning electron microscopy image after switching showing no tearing of the electrode

that[22,23]

$$I_{ISEE} = \beta A e^{-\frac{\varphi_{eff}}{k_B T_e}}. \tag{10}$$

Assuming the barrier height is constant, as might occur if nucleation and growth dominates in the turn-on branch, the two values will be proportional to one another. Consequently, the ISEE signal should obey:

$$I_{ISEE\ turn\ on} \sim \frac{\beta}{\alpha} \sigma, \tag{11}$$

This relationship between the $I_{ISEE}$ and $\sigma$ is unsurprising since it is very similar to the well-studied relationship between the turn-on switching compliance current, device conductance, and filament area, which are also all thought to be proportional[52]. If, however, the effective barrier height varies due to barrier lowering, then the scaling between $I_{ISEE}$ and $\sigma$ can be modeled by:

$$I_{ISEE\ Turn\ on} = \frac{A\beta}{(A\alpha)^{\frac{T_0}{T_e}}} \sigma^{\frac{T_0}{T_e}}, \tag{12}$$

where $\frac{T_0}{T_e}$ is the exponent in a power law scaling between the ISEE signal and the conductivity. Since the hot electron temperature is greater than ambient, $\frac{T_0}{T_e}$ should be strictly less than unity. The extracted value of 0.43 is consistent with this, suggesting an average hot electron temperature of 700 K. While this is a reasonable value for $T_e$, other models are possible and $T_e$ may depend on the model chosen. Understanding the scaling

between fit parameters and values extracted from other methods, such as the temperature coefficient of resistivity of the device, may help clarify these underlying mechanisms[53].

**Imaging the effects of leakage currents around the filament.** Comparing an initial state of the device (Fig. 9a, b) after a stress test (Fig. 9c) to a subsequent device state (Fig. 9d) can provide insights on the distribution of current and power during switching. In the area surrounding the filament, we observe a surrounding dark contrast, most likely crystallized region, which appears to act as a non-programmable leakage path through which excess current can flow (Fig. 9e). Forward biasing the junction during turn off leads to significant power dissipation, with a maximum occurring at −1.7 V and a leakage current of 2 mA (Fig. 9c). Device cycling can lead to electrode coarsening in the crystallized region, and this was observed after a 10-cycle test (Fig. 9c). The SEM images before (Fig. 9a) and after (Fig. 9d) reveal increased secondary electron emission, caused by grain coarsening, in the regions corresponding to the leakage[33–36,54]. It is important to note that power must be dissipating in this region, since electrical connectivity is a precondition for imaging in EBIC, and so the enhanced secondary electron emission is not a sign of the electrode delaminating, which would cause the signal to vanish or share similar magnitude to its immediate neighbors, from where carriers can diffuse. This correlation of power dissipation with the crystallization region, as opposed to being

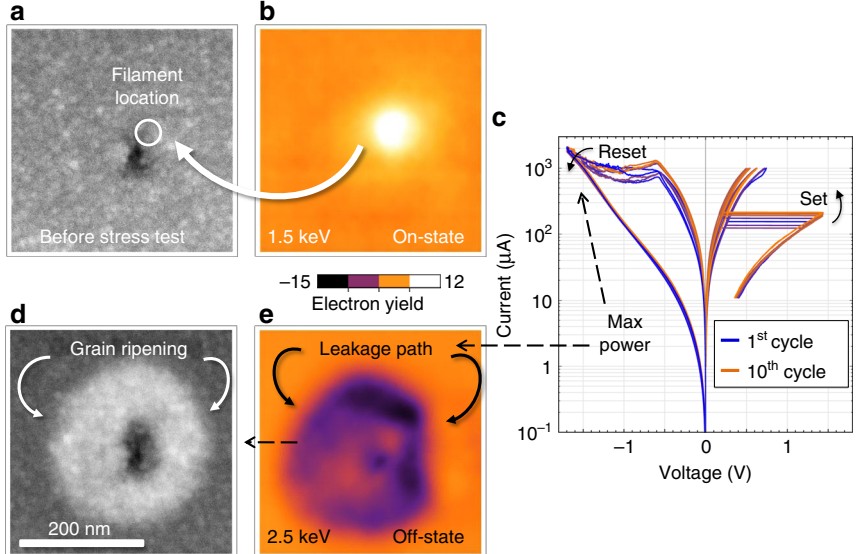

**Fig. 9** Stress testing of device. **a** Scanning electron micrograph of an asymmetric standard device after forming and switching. Area adjacent to the filament is damaged by the switching. **b** Electron beam induced current (EBIC) image of the device in the on state with a low 1.5 keV beam energy. **c** Depiction of a 10 switching curve stress test taken after image a. **d** Scanning electron micrograph and **e** corresponding EBIC image showing correspondence between grain coarsening (seen in **d**) and background crystallization region (region of negative current in e)

centered on the filament, suggests that managing damage induced by the filament formation is more important than controlling changes to the filament itself. Reducing the device size below the breadth of the crystallization region is the simplest possible means of reducing the leakage and excess power dissipation.

With the current density in this region running at ~$7.8 \cdot 10^{10}$ A m$^{-2}$ during the turn off process, a reduction to a 10 nm × 10 nm structure would reduce the leakage to 6 µA. Adding interfacial layers or otherwise engineering the device could also yield benefits by changing the specific contact resistance. While current limiters, such as integrated transistors or ultra-fast pulses, are commonly used to manage current overshoot and device damage, passive means to mitigate the effects of overheating the surrounding device region may be critical for some applications, like passive crossbar arrays[1,55,56].

**Switched device symmetric structure**. The questions of scaling and polarization take on new meaning in a symmetric device with dual Pt electrodes. In the asymmetric structures, overdriving the devices leads to electrode degradation and migration of the switching spot to a new location, however the underlying switching characteristics do not change. Degradation is also present in symmetric structures, but coexists with domains of programmable polarization. This was observed by increasing the voltage stress and cycling the bias between negative and positive polarities, which led to increasing amplitude changes in the orientation of the built-in electric fields as measured by EBIC (Fig. 10a–d and j).

A stepwise motion through the switching reversal (Fig. 10e–i) reveals a propagating domain wall. A measure of the total integrated signal through this transition shows a maximum in the conductivity as the signal sum passes through zero, suggesting highest conductivity when the two polarizations are in balance, with a large domain wall between them (Fig. 10k). The theory of complementary resistive switching suggests that the conductivity will be highest at the interface between these two regions, and so the power dissipation and switching will be preferentially located here (Fig. 10h, l) inducing its propagation[42,46,57]. We observed that some boundaries were less mobile than the primary one,

which may be due to local variations in the grain orientation, crystallinity, or in the composition induced by the high stress.

The differences between the symmetric and asymmetric structures can be attributed to the competing scaling relationships that characterize them. As an asymmetric device is further polarized, its conductivity will only increase until some physical limit, like the temperature of melting, is reached. This is also true in the case of symmetric structures, but a sufficiently polarized device will ultimately decrease in conductivity due to field reversal. The reduction in dissipated power provides an opportunity for adjacent regions to also switch and likewise undergo inversion without the total power dissipation becoming large. This suggests that changes to the device structure, such as the asymmetry, specific resistance, or the heat dissipation, change the scaling of the switching.

**Discussion**

Understanding the physics of EBIC imaging in resistive switching devices has broader implications for the metrology of resistive switching. The direct observation of hot electron currents opens the door to other hot electron techniques such as ballistic electron emission microscopy (BEEM) and internal photoemission (IPE)[58–60]. IPE, being the optical analog to EBIC, sacrifices spatial resolution for precise spectroscopic information. While IPE has long been used to study MIM diodes and other electronic devices[59,61], its use has not been demonstrated on filaments, probably due to the small active area. An IPE system, most likely combined with high brightness sources, focusing optics and phase sensitive detection, could make it possible to deconstruct the underlying electronic structure of the filament–electrode interface as a function of state.

Though EBIC is clearly applicable for conventional device geometries, our results also show its applicability for other geometries, like lateral devices. If the devices studied here were rotated on their side, with EBIC it would be possible to probe the device depletion region at its interface, as in conventional EBIC, and also determine the onset of filament formation by observing the emergence of ISEE at the electrodes. With a lateral device, it would be easier to see structural changes (such as with electron

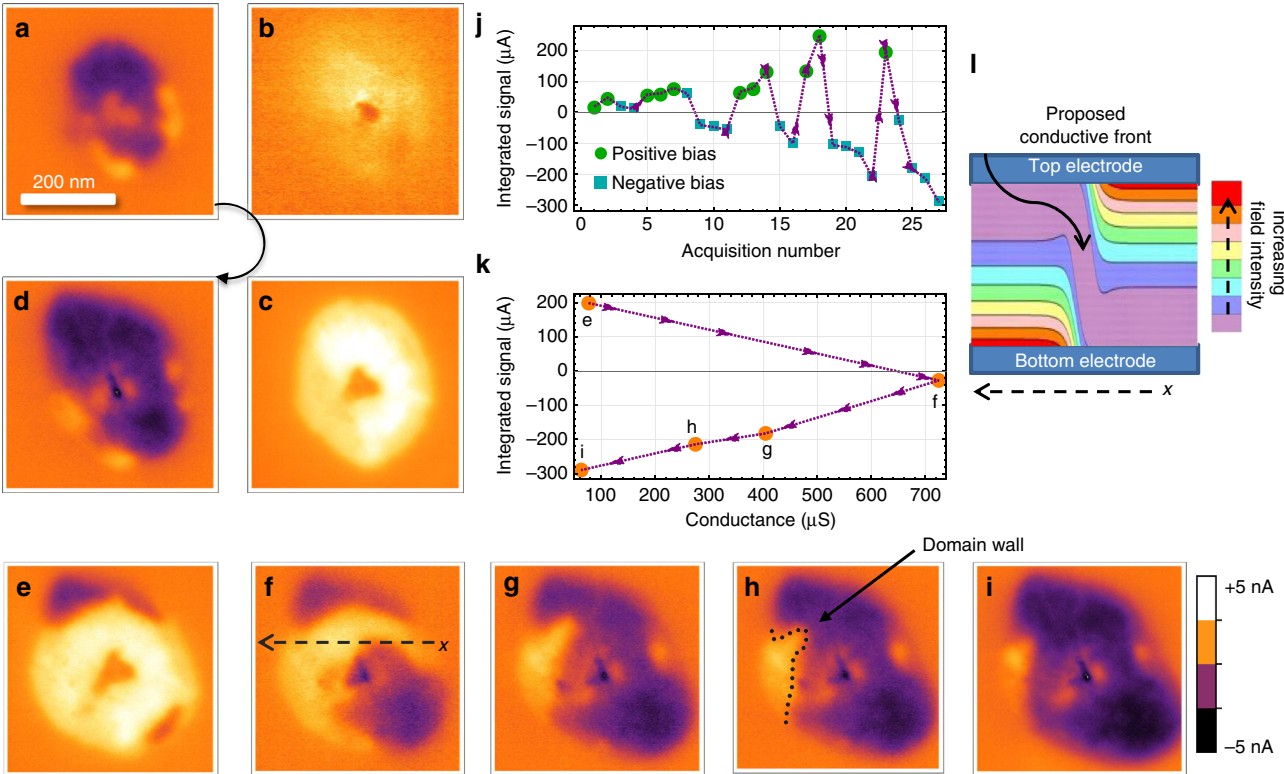

**Fig. 10** Reversible electron–hole signals in symmetric device. **a–d** 5 keV electron beam-induced current (EBIC) images of alternating hard changes in polarization observed in symmetric devices. The sign of the measured current is changing since the device built-in field is changing direction as a consequence of programming. **e–i** Gradual movement of a domain wall around a barrier to encompass the entire switching region. **j** Plot of effect of programming bias on device polarization with increasing amplitude of bias. **k** Relationship between total EBIC signal and conductance (at 0.1 V) from stepwise transition from **e** to **i**. **l** Cartoon depiction of the polarization domains with a reversal in field concentration from the top electrode to the bottom electrode with distance

back scattering diffraction or X-ray absorption) as the device switches, but this could also lead to false positives as regions unrelated to the switching are changed by Joule heating. This problem is particularly acute at large biases where leakage currents could dominate the electrical properties, as seen in Fig. 9. EBIC then can be an effective, rapid means of disentangling resistive switching from artifacts.

We demonstrate energy-dependent and stateful EBIC measurements on conventional resistive switching devices. Comparing these measurements to Monte Carlo simulations reveals two competing forms of contrast that have not previously been distinguished: classic electron–hole pair separation, and ISEE. Differentiating between these two forms of current generation makes it possible to distinguish the filament from its surrounding recrystallized region. Stateful measurements of the ISEE current show different scaling relationships for the turn-on and turn-off branches, which suggests the existence of different, hysteretic mechanisms for filament formation and dissolution. Symmetric device structures show propagating fronts of different polarizations, depending on the direction of the applied bias prior to the image acquisition. This large area switching suggests that the details of device manufacture and geometry can have a significant effect on the underlying scaling of the resistive switching. These effects are difficult to observe spectroscopically, but become clear with EBIC.

## Methods

**Electrical setup and current measurements.** Samples were mounted in a conventional Schottky emission SEM with electrical feedthroughs connected to the device and a stage-mounted Faraday cup for calibrating the injected current. Measurements were done in cycles of grounding the device, programming the

device with a source-measure unit, grounding the device, connecting the current amplifier, and then imaging the device. The effect of changing the device conductance (as measured at 0.1 V) was probed by observing the EBIC signal at individual locations on the device as well as by summing the total signal within an image after subtracting the background due to the surrounding pristine area. Comparing the change in total integrated EBIC signal with respect to the off-state compactly quantifies changes in the state of the device. The image formation mechanisms were probed by imaging the same locations repeatedly with beam energies from 250 eV to 25 keV and then measuring the beam current for each. Plotted ratios of injected beam current to EBIC current (the electron yield, $Y_{EBIC}$) were compared to layer-by-layer energy absorption plots predicted from Monte Carlo simulations of low-energy electrons in the different device structures to determine the origins of different currents[28,62].

**Image processing.** Extracted images are sensitive to effects such as beam–device interactions, 60 Hz noise, device noise, and current–amplifier drift. To minimize these effects, the images were processed using Fourier masking and mean-line leveling to minimize noise and data acquisition artifacts. The images were aligned by doing least-squares minimization of the SEM images and their image off-sets. More information is available in Supplementary Note 2.

**Monte Carlo simulations.** These simulations took as inputs conventionally available values of the density for the materials used as well as independently calibrated film thicknesses from the device fabrication (see Supplementary Note 3). Ten thousand electron trajectories were averaged at a given energy (from 250 eV to 25 keV) to produce a three-dimensional model of the energy absorbed in the device. 2D and 1D plots were generated by numerically integrating all the energy in an individual voxel to produce plots with units of keV nm$^{-2}$ and keV nm$^{-1}$, respectively.

**Device fabrication.** The devices were fabricated with a combination of sputtering and e-beam evaporation. Pt bottom electrodes were sputtered and patterned by Ar ion milling. Subsequently TiO$_x$ was reactively sputtered with an in situ top electrode[63]. The top electrode was ion milled in a mixture of Ar and O$_2$. SiO$_2$ isolation was patterned by liftoff, with undercut providing a gentle slope. The top electrode

contact and subsequent large contacts were deposited by e-beam evaporation of Ti/Au. More details are available in Supplementary Note 1.

**Forming process.** In addition to conventional voltage-induced forming, we also used beam-induced defect formation to improve the reliability of the forming process[64,65]. We found that the dielectric breakdown needed to create switching could be initiated by the combined application of voltage (either current sweeps or voltage pulses) and a large e-beam current at 5 keV. This electron beam assisted forming process made it possible to deterministically locate the breakdown region and consequently the filament (see Supplementary Note 5). A 5 kΩ series resistor was used to limit the current.

**Data availability.** All data are available on reasonable request of the corresponding author. A summary of many important IV curves and images is available in Supplementary Note 7.

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

## Acknowledgements

We thank Brian Thibeault and the UCSB nanofab staff for support in the device development. We thank Alan Band, Glenn Holland, and David Rutter for their support in development of the experimental setup. We thank J. Alexander Liddle for use of his microscope and useful discussions. We thank Andrea Centrone for helpful discussions on the device structure. We thank Mirko Prezioso for helpful discussions on the physics of the measurement. We thank Jason E. Douglas, David Nminibapiel, Mark Stiles, and Alice Mizrahi for reading the manuscript. This work was supported by the AFOSR MURI Grant FA9550-12-1-003 and NIST 70NANB14H185. E.S. acknowledges support under the Cooperative Research Agreement between the University of Maryland and the National Institute of Standards and Technology Center for Nanoscale Science and Technology, Award 70NANB14H209, through the University of Maryland.

## Author contributions

N.Z., A.K., D.B.S., B.D.H., and J.J.M. conceived the experiment. B.D.H., G.C.A., and D.B.S. designed the devices. B.D.H. and G.C.A. fabricated the devices. B.D.H., E.S., A.K., N.Z., and J.J.M. developed experimental procedures and designed electronics. B.D.H. performed the measurements and wrote the manuscript. All contributors discussed the experimental results and edited the manuscript.

## Additional information

**Competing interests:** The authors declare no competing financial interests.

