## [Peer Review File · Nature Communications]

Reviewers' Comments:

Reviewer #1:

Remarks to the Author:

This manuscript reports the characterization and understanding of TiO₂ based resistive switching devices using electron beam induced current (EBIC) as a probe to resistance states of the devices. Three different cell structures, varied in terms of electrode symmetry and stacking sequence, were investigated, and the spatial distribution and summed EBIC intensity were utilized to correlate with the processes of filament formation and resistive switching. The topic of resistive switching materials and device physics is of principle interest, however, I have the following comments and concerns as for the significance and technical aspects of this work, which in my view precludes it from publication in Nature Communications:

1. The primary issue with this study is that EBIC is not a newly developed technique, nor is this the first time for the approach to be used for the understanding of filament formation and resistive switching in memristive devices. In fact, a quite substantial part of the experimental findings in this work, in particular the observation of conducting and defect spots in EBIC images as well as the morphological changes after switching, have been discussed and understood in previous studies such as Refs. 25 and 26. This generally eats into the novelty of the present submission and makes it more suitable for a specialized journal.

2. To demonstrate the proposed approach as a reliable and effective one for stateful characterization of resistive switching devices, it seems to me very important to testify the main body of the conclusions in a diversity of material and device systems, e.g. other memristive oxides or ideally in CBRAM systems as well. Such investigations and discussions shall be favorable for assessing the general applicability of the proposed technique.

3. It is well known that exposure of specimens to electron beam in SEM is likely to result in deposition of carbon based contaminations on sample surfaces. Will this impose any influences on the EBIC measurements, as this was performed in SEM?

4. The interpretation of EBIC results largely relies on Monte Carlo simulations in this work. The validity of the model is therefore critical for correctly understanding the data and assigning the origin of the different components in EBIC signals. It may thus be useful to shed light on the construction of the model in more detail and discuss how the correctness of the simulation can be verified.

5. The existence of conducting filaments manifests as a spot in the experimental setup, as the imaging was performed along the vertical direction. Will the measurements will be applicable to lateral resistive switching devices as well, so that more structural information of the filaments can be disclosed? Please comment.

6. A minor comment is that the stacking sequence was not described in a consistent manner in the manuscript, as both top-to-bottom and bottom-to-top sequences were used. This should be kept consistent throughout the manuscript.

Reviewer #2:

Remarks to the Author:

Authors report on the observation of the filament in working resistive switching devices by Electron Beam Induced Current technique and interpret the changes induced by the application of bias. The observations are novel, very intriguing, and of potentially significant impact.

However, I cannot recommend the publication in the present form. Part of the problems stem from

the writing style: the paper is not well written and very difficult to follow. For example, on page 2, authors state: "This ion motion leads to a local, nanometer-scale variation in the vacancy concentration and a corresponding variation in the thickness of the oxide's depletion region." Clearly, authors have a specific model of resistive switching in mind but it is not clear what it is. Most of the resistive switching models do not invoke Schottky barriers or modulation of their height during switching. It would help if authors clearly described the model they are using and included the references to the original work. The new elements of the model should be clearly stated and supported by the experimental evidence. Problems of this type are numerous.

A bigger issue stems from a comment on page 7: "Switched-on devices exhibited morphological changes, namely local electrode reconstruction (both minor and severe)". Authors also refer to this area as "deformation".

This effect is common in resistive switching devices and is typically associated with excessive current flow during electroformation and/or switching. Authors correctly mention that the presence of this artifact makes it very difficult to analyze the changes associated with resistive switching. Unfortunately, this statement also applies to this report. Authors have not identified what is the mechanism for this contrast in EBIC images. It is not clear what is it due to or how to differentiate the filament from the deformation (EBIC is measuring only the current induced by the beam not the current at a given location). For example, Fig. 6 shows a large area of EBIC contrast with three features mentioned by authors. There is a filament (circled in the figure) which is not readily visible and is not clear how authors have it identified. The dark area next to the filament is dark in all micrographs and is denoted as "damage". Again, it is not clear how authors came to this conclusion. The third area is described as "grain ripening". The term is not explained, reference 39 does not use this term, and gives little insight into what it could be.

While the damage to device structures is common, it is by no means unavoidable. Authors could either improve the testing procedures by controlling the current flow or redesign the devices used in the experiment.

In summary, I view the results as worthy of publication and potentially very significant. I would encourage authors to address the problems and resubmit the paper.

Reviewer #3:

Remarks to the Author:

The paper deals with the ReRAM filamentary resistive switching observed with EBIC technique. Although EBIC is not new, the experimental work is sufficiently thorough to provide images of the filament evolution during resistive switching with a much better resolution than the state of the art. The aforementioned technique is very interesting for ReRAM operation understanding and maybe also for practical device optimization. It will interest many laboratories involved in the field. The technique is also used to analyze the differences on filament EBIC images between symmetric and asymmetric structures and provide a novel interpretation. The paper is clear and well organized. I think that it is then acceptable for publication in Nature communication. However before publication I recommend that the following points should be addressed.

- 1) One can expect that the conducting filament in a ReRAM is not a perfect cylinder but it likely presents different section area along its length. Thus the filament current is certainly limited by the part of the filament with the least section area. At the location of this least section area one expects the largest current density as power dissipation. Now the EBIC images the filament current from the top. Is it possible to quantify the effect of this constriction location along the filament on the EBIC measurements? What does happen if these constrictions are in the vicinity of top or bottom electrodes?

2) In section III, // the case of asymmetric structures is discussed. It is not clear to me if it is asymmetric standard or asymmetric inverted. Do the authors observe some differences in EBIC characterization between both asymmetric standard and inverted devices?

3) In page 9 (the first and second lines) the authors introduce and use later the assumed Arrhenius variation of conductance with temperature. However it has been shown experimentally for bipolar filamentary switching that on state as off state conductions are temperature independent from 4K to 300K [1]. Can the authors comment?

4) I'm not sure to understand completely the argument in equation 6. It seems to me that equation 6 is the result of combining equations 5 and 4 without any assumption on barrier height whatever it is a variable or a constant. May be that equation 4 stands only for the turn off switching? Assuming that on branch conduction is simply proportional to the filament area (as for example Ohm's law, Sharvin conductance or quantum point contact conductance) might be enough to explain the observations. This will be also consistent with the author's conclusions regarding the differences between on branch and off branch.

5) In figure 5 b) left and page 10 lines 5 and 6 It is showed that I_{see} is proportional to the conductance $I_{see} = V_s \cdot \sigma$ (σ being the conductance) and $V_s = 0.1V$. Do the authors think that this relation can be compared with the well-known law on resistive filamentary based ReRAM $I_c = V_c \cdot \sigma$ [2]? (I_c being the compliance current imposed by an external device during the positive set voltage ramp and $V_c \sim 0.4V$). Note that in [1] the V_c value was calculated on a filament area growth basis, which is also the assumption made by the authors.

6) The current-voltage characteristic for asymmetric structure (figure 6 b) corresponds to asymmetric standard structure? The authors may also, for sake of comparison, show the corresponding switching characteristics for the symmetric structure in figure 7 for example.

[1] S. Blonkowski, T. Cabout, J. Phys. D 48,345101 (2015)

[2] D. Ielmini, IEEE Trans. Electron. Dev. VOL. 58, NO. 12, (2011)

Response to Reviewer Comments

The authors would like to thank all reviewers for reading the manuscript so carefully. The many useful suggestions are greatly appreciated. In response, we have revised the manuscript thoroughly, and believe that these changes have improved our paper substantially. This documents consists of two parts

- 1) A brief list of the most significant changes and new material in the manuscript, and
- 2) a detailed response to all of the referees comments which also reviewer all of the major changes in the text.

1) Summary of Major Changes

- a) To increase clarity of the presentation, the number of figures has been increased from 7 to 10 by:
 - I. Splitting up the EBIC energy dependent imaging figures into the micrographs and the spectra separately.
 - II. Adding a new figure detailing the radial profiles of the switching spot.
- b) In the energy dependent imaging figures all of the SEM images are displayed more prominently to show that there is no electrode tearing.
- c) The color scheme for the EBIC image has been updated in the main text.
- d) In Figure 9 (old figure 6), the means of determining the filament location are clarified.
- e) The energy dependent imaging experiment was repeated using a pulse generator to form the device, reducing changes to the electrode and the size of the surrounding recrystallized region. This did not result in any changes to our conclusions however. The old data was moved to the supplement.
- f) To increase clarity, the beam energy dependent EBIC spectra are taken from the on-state and the off-state instead of from the filament and the surrounding region in the same state.
- g) The energy dependent EBIC yields were quantitatively compared to the Monte Carlo simulation data by doing a least-squares fit of the modeled energy absorption. The top Al_2O_3 was added to the definition of the top electrode to get an appropriate fit at the lowest beam energies.
- h) Additional information was added about the Monte Carlo modeling to the main text as well as the supplement.

- i) Commentary on the plasma cleaning procedure to prevent carbon contamination was added.
- j) A new section was added at the end putting our observations concerning hot electron transport into a broader scientific and metrological context.
- k) Additional references were added to justify the explanation that the increase in secondary electron emission in Figure 9 (old Figure 6) is due to an increase in surface roughness and grain coarsening.
- l) Additional references were added to justify the explanation that the increase in the dark signal is due to grain coarsening. The device “damage” has been recharacterized as proposed “recrystallization” in the TiO₂ to distinguish it from electrode damage or degradation of the top Pt.
- m) Additional references were added to justify the use of a Schottky barrier model for the particular case of TiO₂ and the section was re-written to be more specific to this model.
- n) Additional commentary and references were added to comment on the relationship between the internal secondary electron emission, the on-state conductance, and the temperature dependence. The limitations of the model are also stressed.
- o) Additional commentary and references were added to discuss the relationship between internal secondary electron emission, the switching compliance current, and the device conductance.
- p) Various other small typos were corrected. Important changes are highlighted in a separately attached document.

2) Referee Response

Since each of the referees had very different primary points of contention, we have decided to address each reviewer individually.

Reviewer #1 (Remarks to the Author):

This manuscript reports the characterization and understanding of TiO₂ based resistive switching devices using electron beam induced current (EBIC) as a probe to resistance states of the devices. Three different cell structures, varied in terms of electrode symmetry and stacking sequence, were investigated, and the spatial distribution and summed EBIC intensity were utilized to correlate with the processes of filament formation and resistive switching. The topic of resistive switching materials and device physics is of principle interest, however, I have the following comments and concerns as for the significance and technical aspects of this work, which in my view precludes it from publication in Nature Communications:

1. The primary issue with this study is that EBIC is not a newly developed technique, nor is this the first time for the approach to be used for the understanding of filament formation and resistive switching in memristive devices. In fact, a quite substantial part of the experimental findings in this work, in particular the observation of conducting and defect spots in EBIC images as well as the morphological changes after switching, have been discussed and understood in previous studies such as Refs. 25 and 26. This generally eats into the novelty of the present submission and makes it more suitable for a specialized journal.

We agree EBIC has been done before, but we also think there is a substantial difference between getting a picture and doing a spectroscopic and stateful investigation of resistive switching. Our investigation provides mechanistic information about the image formation process, especially with respect to internal secondary electron emission in the devices. No such physical information has been provided previously, either qualitatively or quantitatively. The images, therefore, were difficult or impossible to interpret. Also, our improved resolution, a result of better device design and a superior microscope, make it possible to look at the switching spot fine structure.

It is worthwhile noting that the situation of our work is quite analogous to other similar published studies. For example Baeumer, *et al.*[1] and Kumar[2] published results on single devices while previous x-ray studies had already been performed by Strachan, *et al.*[3] The distinction was that the quality of the experiments has allowed entirely new information to be learned, even if the experimental tools are not brand new. I would add that in our work, we can put a substantially higher degree of confidence on the filament location and have performed enough measurements to even generate animations of the switching process. We have provided one such animation in the supplement.

Observing internal secondary electron emission has significant experimental and theoretical implications. We have added a new section of the paper commenting on this to address the question of novelty as well as to address comment number 5.

2. To demonstrate the proposed approach as a reliable and effective one for stateful characterization of resistive switching devices, it seems to me very important to testify the main body of the conclusions in a diversity of material and device systems, e.g. other memristive oxides or ideally in CBRAM systems as well. Such investigations and discussions shall be favorable for assessing the general applicability of the proposed technique.

We would agree with this comment if we had conducted the study of Ref. 25; however the focus of our investigation is not simply on producing images but rather on providing spectroscopic, energy, and state dependent data on the device response to the incident beam current. To date, no such study has been performed.

Conducting more experiments on more devices is a good idea and we plan to do more investigations, but these investigations will undoubtedly generate new and interesting information about those materials systems which should be in their own publications.

3. It is well known that exposure of specimens to electron beam in SEM is likely to result in deposition of carbon based contaminations on sample surfaces. Will this impose any influences on the EBIC measurements, as this was performed in SEM?

This is a very important point. We have taken care to ensure that there is no carbon deposition by doing 20 minute oxygen plasma cleans on the samples after they have been loaded into the chamber. We then ran a standardized test to ensure that for the duration of the experiment no carbon was deposited. We have recorded this information and it is in the supplement and we have added information in the main text addressing it to make it more clear.

4. The interpretation of EBIC results largely relies on Monte Carlo simulations in this work. The validity of the model is therefore critical for correctly understanding the data and assigning the origin of the different components in EBIC signals. It may thus be useful to shed light on the construction of the model in more detail and discuss how the correctness of the simulation can be verified.

We have added more information into the main text as well as expanded the supplement. We base the correctness of the model on its ability to predict the locations of the maximum and minimum currents. We have also improved our analysis by including a least-squares fit of the energy absorption data to the EBIC data. In this model, the energy absorption curves from the simulations serve as a basis set to linearly reconstruct the measured data.

The magnitude of an EBIC signal is known to conventionally depend on both the location of the deposited energy as well as the amount of deposited energy. Our model captures the most important features, but we believe only a more detailed transport model of hot electrons could perfectly recreate the data. It is worth noting that we have not done any optimization of the simulations to better fit the data and we use only the independently measured fabrication parameters from the device manufacturing in order to derive our energy absorption curve basis set.

To update the simulations we have included the energy absorbed into the Al_2O_3 layer in the top electrode absorption measurement. This was to better fit the low energy absorption curve of the on state data. We believe this is justified since the full width half maximum at low energy is much larger than the film thickness.

5. The existence of conducting filaments manifests as a spot in the experimental setup, as the imaging was performed along the vertical direction. Will the measurements will be applicable to lateral resistive switching devices as well, so that more structural information of the filaments can be disclosed? Please comment.

The EBIC signal would in fact be sensitive in the lateral direction. An absolutely critical feature to point out here is that the EBIC signal would be most sensitive to the region where the field is most rapidly changing, e.g., the point of resistive switching. This is one of the main regions why our study is important. Since we analyze all the different contributions to the EBIC current (electron-hole pair separation and hot electron transmission), we provide a framework for using it quantitatively and qualitatively. If such a lateral study were conducted without EBIC, it might not be possible to disentangle changes in the switching with random artifacts in the measurement. For example, a lateral switch could have crystallization occur over a very wide range, and this would be visible by an enhanced electron-hole pair separation current at the device body. However, if the device were a Schottky barrier type switch, as the device switched on, one would expect to see the depletion width of the electron-hole pair region shrink at the location of the filament but the intensity increase as total recombination declined due to the shorter diffusion distance. As the device turned fully on, excitation of the electrode would begin to cause efficient injection of charge across the interface, producing a precise location for the switching filament and expansion of the measured EBIC signal across the film-electrode barrier. The signal to noise ratio would be very favorable since it's possible to inject the entire beam into a part of a lateral device by operating it at low beam energy. There would be no ambiguity since ratios of EBIC current to beam current are as high as 20 at the filament location.

One component of our long-term plans is to characterize a device at beam energies from 100 eV to 100 keV, thereby analyzing both its electronic and structural components *in situ* using the broadest dynamic range of incident electron beam energies. The current study is the first step whereby the precise interaction between a ReRAM device and a low incident beam energy are understood. Such a study would involve substantially more device engineering than has been presently done.

To address the reviewer's comment in the manuscript, we have added a new section at the end of the paper in which we discuss the above points.

6. A minor comment is that the stacking sequence was not described in a consistent manner in the manuscript, as both top-to-bottom and bottom-to-top sequences were used. This should be kept consistent throughout the manuscript.

We have edited the manuscript to maintain consistency.

Reviewer #2 (Remarks to the Author):

Authors report on the observation of the filament in working resistive switching devices by Electron Beam Induced Current technique and interpret the changes induced by the application of bias. The observations are novel, very intriguing, and of potentially significant impact.

We thank the reviewer for the positive commentary. We believe specifically our contribution about hot electron transport near the filament and increased electron-hole pair separation in the surrounding region to be of most significance and important as well.

However, i cannot recommend the publication in the present form. Part of the problems stem from the writing style: the paper is not well written and very difficult to follow.

We have gone through the manuscript and rewritten large parts of it. We have also decided to use more of the length allowed to be clear.

For example, on page 2, authors state: "This ion motion leads to a local, nanometer-scale variation in the vacancy concentration and a corresponding variation in the thickness of the oxide's depletion region." Clearly, authors have a specific model of resistive switching in mind but it is not clear what it is. Most of the resistive switching models do not invoke Schottky barriers or modulation of their height during switching.

Schottky barrier modulation is a very common model of resistive switching and has been the subject of numerous papers including Yang, *et al.*[4-6], Bauemer, *et al.*[1], Marechewka, *et al.*[7], Hur, *et al.* [8], and Park, *et al.*[9]. We believe the source of the confusion may lie in that the reviewer may be more familiar with CBRAM such as in Hubbard, *et al.*[10], where Schottky barrier modulation is not relevant. In CBRAM there is a clear separation of the phases including a phase boundary. In oxide resistive switches, such phase boundaries may not exist or be as sharp. Consequently the physics of the conductivity changes can be different.

A glance through the literature would suggest that Schottky models are more commonly applied to oxides with higher dielectric constants (such as TiO_2 and SrTiO_3) whereas other models, such as hopping, are more common for devices with a more intermediate dielectric constant (HfO_2 and Ta_2O_5) or low dielectric constant (SiO_2).

We have re-written parts of these sections to increase clarity.

It would help if authors clearly described the model they are using and included the references to the original work. The new elements of the model should be clearly stated and supported by the experimental evidence. Problems of this type are numerous.

We have added some clarification including citations as to why we think TiO_2 is a Schottky barrier type switch. The strongest evidence however comes from the literature as well as the I-V curves which show a marked asymmetry. This can be seen in Figure 9, where we show I-V curves. Negative applied currents (forward bias on the bottom electrode) are much higher in the off state. The EBIC electron-hole pair separation current itself, such as shown in Figure 2, is further support for the presence of a built in field across the device.

A bigger issue stems from a comment on page 7: "Switched-on devices exhibited morphological changes, namely local electrode reconstruction (both minor and severe)". Authors also refer to this area as "deformation".

This effect is common in resistive switching devices and is typically associated with excessive current flow during electroformation and/or switching. Authors correctly mention that the presence of this artifact makes it very difficult to analyze the changes associated with resistive switching. Unfortunately, this statement also applies to this report. Authors have not identified what is the mechanism for this contrast in EBIC images.

We believe we have not been fully clear with the magnitude or degree of damage in each and every case as well as how it should be interpreted. For this reason, we have modified all the figures to clearly show the SEM image of the device after the experimental testing has taken place. In figure 7 (Figure 5 in the old version), which had the current as a function of switching state, we have also added an SEM image. There are no visible breaks in the electrode.

In figure 5 (old figure 4), the spectroscopic data of the EIBC current, there were three small holes in the top electrode near the filament (see figure 9 (old figure 6); in other SEM images the tear is larger due to more device cycling). Since they were adjacent to the filament, not on the filament, we did not think there was any influence on the result. Nevertheless, to remove ambiguity, we repeated the experiment and are showing the SEM image of the device with no breaks in the contact. The device state was much more conductive than the previous example, leading to a much higher internal secondary electron (hot electron) current, but the results are the same. There is a positive peak at around 1.5 keV and a negative peak at around 3-5 keV. To make this clearer, we have added a new linear combination model to fit the data and done an on-off comparison as opposed to a filament-neighboring region comparison. We further believe that the strong correspondence between the theory and the experiment are evidence of both the correctness of our model and the intact structure of our device. The original spectroscopic

EBIC data is now in the supplement and its electrode SEM image after the measurement is available there.

For the stress testing experiment there was significant electrode damage, but this is also the point of the experiment, so we have left this figure intact. We also have more commentary on some of the effects visible on the electrode. This also addresses the following point.

It is not clear what is it due to or how to differentiate the filament from the deformation (EBIC is measuring only the current induced by the beam not the current at a given location). For example, Fig. 6 shows a large area of EBIC contrast with three features mentioned by authors. There is a filament (circled in the figure) which is not readily visible and is not clear how authors have it identified. The dark area next to the filament is dark in all micrographs and is denoted as "damage".

We have updated Figure 9 (old figure 6) to more clearly demonstrate how, even with a damaged device, it is possible to determine the precise filament location. We are showing the device in the on state with a low beam voltage, 1.5 keV. Based on the spectral data shown in Figure 5, the filament should produce a maximum signal and there should be no negative signal from the TiO₂ layer. The filament location is clearly visible at this location independent of all other features. This is because the imaging mechanism is fundamentally sensitive to changes in the electronic conductivity and, more importantly, only those changes to the electronic conductivity which are physically connected to the programming and current amplifier circuitry, further minimizing the potential for confusion.

We also make comparisons to the same device in the off state but with 2.5 keV. At this energy there should be significant TiO₂ layer absorption. We believe the increased dark current in this region is due to crystallization of the TiO₂. After power cycling the device, grain coarsening of the top electrode appears to correlate with this region, not with the filament location.

Again, it is not clear how authors came to this conclusion.

The third area is described as "grain ripening".

The term is not explained, reference 39 does not use this term, and gives little insight into what it could be.

We apologize for the mixed up definitions. Reference 39 does not say "grain ripening", instead it uses the term "grain coarsening." We have adjusted our terminology to be consistent. We have also added a reference on grain ripening and coarsening in nanostructured platinum[11].

Grain coarsening occurs when a material is subjected to high temperatures. It can cause small grains to shrink, large ones to grow, as well as result in increased surface roughness. Increased

surface roughness is known to correlate with increased secondary electron emission, and we have added references supporting this statement[12-14].

While the damage to device structures is common, it is by no means unavoidable. Authors could either improve the testing procedures by controlling the current flow or redesign the devices used in the experiment.

In summary, i view the results as worthy of publication and potentially very significant. I would encourage authors to address the problems and resubmit the paper.

To make the damage as small as possible for our duplication of the spectroscopic measurement in Figure 5, we used a pulse generator to do the device forming. While we could have used our older procedure and not had electrode damage, this method produced a surrounding dark region that was smaller and more symmetric. This allowed us to make a new figure, Figure 6, around which we did azimuthal integration. The choice of a much more conductive state for the spectroscopy also made a previously unseen feature visible, i.e., the internal secondary electron emission from the bottom electrode to the top electrode. This is visible in Figure 7. We believe these changes address the author's primary concern as well as add more interesting information to the paper.

Reviewer #3 (Remarks to the Author):

The paper deals with the ReRAM filamentary resistive switching observed with EBIC technique. Although EBIC is not new, the experimental work is sufficiently thorough to provide images of the filament evolution during resistive switching with a much better resolution than the state of the art. The aforementioned technique is very interesting for ReRAM operation understanding and maybe also for practical device optimization. It will interest many laboratories involved in the field. The technique is also used to analyze the differences on filament EBIC images between symmetric and asymmetric structures and provide a novel interpretation. The paper is clear and well organized. I think that it is then acceptable for publication in Nature communication. However before publication I recommend that the following points should be addressed.

We appreciate the reviewer's positive commentary.

1) One can expect that the conducting filament in a ReRAM is not a perfect cylinder but it likely

presents different section area along its length. Thus the filament current is certainly limited by the part of the filament with the least section area. At the location of this least section area one expects the largest current density as power dissipation. Now the EBIC images the filament current from the top. Is it possible to quantify the effect of this constriction location along the filament on the EBIC measurements? What does it depend on if this constriction is in the vicinity of top or bottom electrodes?

We have considered this very carefully and we think that the most interesting way to determine the device depth would be to analyze the scaling dependence of the internal secondary electron current in detail. The hot electron transmission through the oxide will have some characteristic mean free path and collection efficiency, which we think would be maximally sensitive to distance from the heat source. Consequently, a filament closer to the surface would probably be more easily visible and have a higher signal magnitude. Such effects would be interesting to consider in a future study with a more comprehensive model of the electron transmission specifically for the case of a filament. Such studies have already been done, but not for the case of resistive switches.

We have made sure to re-emphasize that our model is a first order model and not comprehensive in the paper.

2) In section III, II the case of asymmetric structures is discussed. It is not clear to me if it is asymmetric standard or asymmetric inverted. Do the authors observe some differences in EBIC characterization between both asymmetric standard and inverted devices?

We have clarified the point. We have data about the inverted structure in the supplement section. We did not investigate it as much since there was only one polarity for the current making it more difficult to distinguish what is occurring. However, using spectral measurements, it is possible to determine everything. We have opted to leave this full characterization for a future study, since the key finding we have reported is that there is only one signal polarity: positive.

3) In page 9 (the first and second lines) the authors introduce and use later the assumed Arrhenius variation of conductance with temperature. However it has been shown experimentally for bipolar filamentary switching that on state as off state conduction are temperature independent from 4K to 300K [1]. Can the authors comment?

We have reviewed the provided reference and we agree it shows that the on-state conductance is temperature independent. We believe this is consistent with our results because in the limit of lower barrier height we would expect the temperature dependence to disappear.

However, the reference does not show explicit temperature independence in the off state. Other experiments suggest that the off state does in fact have a strong temperature dependence[15].

Because of different competing claims as to the resistive switching mechanism, it could also well be that our device (and other TiO_2 based devices) and Blonkowski's device have different properties. We would expect them to have different scaling dependences on the hot electron transmission. However, in the limiting case of the on state, they should be very similar since in this state there is no barrier to conduction whereas in the off state different factors could limit the current.

In the updated section concerning the model of both conductivity and the ISEE formation mechanism, we have noted the limitations of our model in the high temperature limit.

4) I'm not sure to understand completely the argument in equation 6. It seems to me that equation 6 is the result of combining equations 5 and 4 without any assumption on barrier height whatever it is a variable or a constant. May be that equation 4 stands only for the turn off switching? Assuming that on branch conduction is simply proportional to the filament area (as for example Ohm's law, Sharvin conductance or quantum point contact conductance) might be enough to explain the observations. This will be also consistent with the author's conclusions regarding the differences between on branch and off branch.

The reviewer is correct. Equation 6 only applies to the turn off branch. We have modified the paper to make it clearer.

We would like to emphasize that this model may not be the definitive model on the subject. For example, Blonkowski and Cabout (reviewer 3's reference 2) introduced their own variable, P , the contention probability. Such variables could linearize the internal secondary electron emission during turn-off. Further investigation into this approach would indeed be interesting, but we believe is beyond the scope of the current paper since in this work we have only just discovered the internal secondary electron emission current in resistive switches and determined its most basic origin.

5) In figure 5 b) left and page 10 lines 5 and 6 It is showed that I_{see} is proportional to the conductance $I_{see} = V_s \cdot \sigma$ (σ being the conductance) and $V_s = 0.1V$. Do the authors think that this relation can be compared with the well-known law on resistive filamentary based ReRAM $I_c = V_c \cdot \sigma$ [2]? (I_c being the compliance current imposed by an external device during the

positive set voltage ramp and $V_c \sim 0.4V$). Note that in [1] the V_c value was calculated on a filament area growth basis, which is also the assumption made by the authors.

We completely agree with the reviewer and thank him/her for this observation. We have added this into the paper and make this comparison explicitly. We believe the scaling of device resistance with both compliance current and ISEE are complementary observations.

6) The current-voltage characteristic for asymmetric structure (figure 6 b) correspond to asymmetric standard structure? The authors may also, for sake of comparison, show the corresponding switching characteristics for the symmetric structure in figure 7 for example.

We appreciate the reviewer's comment and respectfully note that all of the I-V curves for all the measurements are shown in the supplementary information.

[1] S. Blonkowski, T. Cabout, *J. Phys. D* 48,345101 (2015)

[2] D. Ielmini, *IEEE Trans. Electron. Dev.* VOL. 58, NO. 12, (2011)

- [1] C. Baeumer, *et al.*, "Quantifying redox-induced Schottky barrier variations in memristive devices via in operando spectromicroscopy with graphene electrodes," vol. 7, p. 12398, 2016.
- [2] S. Kumar, *et al.*, "Direct Observation of Localized Radial Oxygen Migration in Functioning Tantalum Oxide Memristors," *Advanced Materials*, vol. 28, pp. 2772-2776, 2016.
- [3] J. P. Strachan, *et al.*, "Structural and chemical characterization of TiO_2 memristive devices by spatially-resolved NEXAFS," *Nanotechnology*, vol. 20, p. 485701, 2009.
- [4] J. J. Yang, *et al.*, "A family of electronically reconfigurable nanodevices," *Advanced Materials*, vol. 21, pp. 3754-3758, 2009.
- [5] J. J. Yang, *et al.*, "Metal/ TiO_2 interfaces for memristive switches," *Applied Physics A*, vol. 102, pp. 785-789, 2011.
- [6] J. J. Yang, *et al.*, "Engineering nonlinearity into memristors for passive crossbar applications," *Applied Physics Letters*, vol. 100, p. 113501, 2012.
- [7] A. Marchewka, *et al.*, "Physical simulation of dynamic resistive switching in metal oxides using a Schottky contact barrier model," in *2015 International Conference on Simulation of Semiconductor Processes and Devices (SISPAD)*, 2015, pp. 297-300.
- [8] J. H. Hur, *et al.*, "Modeling for bipolar resistive memory switching in transition-metal oxides," *Physical Review B*, vol. 82, p. 155321, 2010.
- [9] J. Park, *et al.*, "Multibit Operation of TiO_2 -Based ReRAM by Schottky Barrier Height Engineering," *IEEE Electron Device Letters*, vol. 32, pp. 476-478, 2011.
- [10] W. A. Hubbard, *et al.*, "Nanofilament Formation and Regeneration During Cu/ Al_2O_3 Resistive Memory Switching," *Nano Letters*, vol. 15, pp. 3983-3987, 2015/06/10 2015.
- [11] B. Gao, *et al.*, "In situ transmission electron microscopy imaging of grain growth in a platinum nanobridge induced by electric current annealing," *Nanotechnology*, vol. 22, p. 205705, 2011.

- [12] J. Kawata and K. Ohya, "Surface Roughness Effect on Secondary Electron Emission from Beryllium under Electron Bombardment," *Journal of the Physical Society of Japan*, vol. 63, pp. 795-806, 1994.
- [13] N. Kenji, *et al.*, "Influence of Surface Roughness on Secondary Electron Emission and Electron Backscattering from Metal Surface," *Japanese Journal of Applied Physics*, vol. 33, p. 4727, 1994.
- [14] Y. C. Yong, *et al.*, "Determination of secondary electron yield from insulators due to a low-kV electron beam," *Journal of Applied Physics*, vol. 84, pp. 4543-4548, 1998/10/15 1998.
- [15] C. Walczyk, *et al.*, "Impact of Temperature on the Resistive Switching Behavior of Embedded HfO₂-Based RRAM Devices," *IEEE Transactions on Electron Devices*, vol. 58, pp. 3124-3131, 2011.

Reviewers' Comments:

Reviewer #1:

Remarks to the Author:

The authors have made satisfactory responses/explanations to my questions in the first round of review. The authors have amended the manuscript properly with additional results and discussions, and the revised manuscript has shown substantial improvements compared with the original version. In my opinion the revised manuscript may now be accepted for publication.

Reviewer #2:

Remarks to the Author:

My comments have been addressed adequately and the manuscript can be accepted as is.